# Nutrition Therapy in Critically Ill Patients with Obesity: An Observational Study

**DOI:** 10.3390/nu17040732

**Published:** 2025-02-19

**Authors:** Juan Carlos Lopez-Delgado, Laura Sanchez-Ales, Jose Luis Flordelis-Lasierra, Esther Mor-Marco, M Luisa Bordeje-Laguna, Esther Portugal-Rodriguez, Carol Lorencio-Cardenas, Paula Vera-Artazcoz, Sara Aldunate-Calvo, Beatriz Llorente-Ruiz, Rayden Iglesias-Rodriguez, Diana Monge-Donaire, Juan Francisco Martinez-Carmona, Rosa Gastaldo-Simeón, Lidón Mateu-Campos, Maria Gero-Escapa, Laura Almorin-Gonzalvez, Beatriz Nieto-Martino, Clara Vaquerizo-Alonso, Teodoro Grau-Carmona, Javier Trujillano-Cabello, Lluis Servia-Goixart

**Affiliations:** 1Area de Vigilancia Intensiva, Clinical Institute of Internal Medicine & Dermatology (ICMiD), Hospital Clínic de Barcelona, C/Villarroel, 170, 08036 Barcelona, Spain; lalmorin@clinic.cat; 2Intensive Care Department, Hospital de Terrassa, C/Torrebonica, s/n, 08227 Terrassa, Spain; pelirrotxa@gmail.com; 3Intensive Care Department, Hospital Universitario 12 de Octubre, Av. de Córdoba s/n, 28041 Madrid, Spain; makalyconru@gmail.com (J.L.F.-L.); teograu2@gmail.com (T.G.-C.); 4i+12 (Instituto de Investigación Sanitaria Hospital 12 de Octubre, Research Institute Hospital 12 de Octubre), Av. de Córdoba s/n, 28041 Madrid, Spain; 5Intensive Care Department, Hospital Universitario Germans Trias i Pujol, Carretera de Canyet, s/n, 08916 Badalona, Spain; esthermormarco@gmail.com (E.M.-M.); luisabordeje@gmail.com (M.L.B.-L.); 6Intensive Care Department, Hospital Clínico Universitario de Valladolid, Av. Ramón y Cajal, 3, 47003 Valladolid, Spain; esther_burgos@hotmail.com; 7Intensive Care Department, Hospital Universitari Josep Trueta, Av. de França, s/n, 17007 Girona, Spain; carol_lorencio@hotmail.com; 8Intensive Care Department, Hospital de la Santa Creu i Sant Pau, C/Sant Quintí, 89, 08041 Barcelona, Spain; pvera@santpau.cat; 9Intensive Care Department, Complejo Hospitalario de Navarra, C/Irunlarrea, E, 31008 Pamplona, Spain; saldunate298@hotmail.com; 10Intensive Care Department, Hospital Universitario Príncipe de Asturias, Av. Principal de la Universidad, s/n, 28805 Alcalá de Henares, Spain; bllorenteruiz@gmail.com; 11Intensive Care Department, Hospital General de Granollers, C/Francesc Ribas, s/n, 08402 Granollers, Spain; riglesias@fhag.es; 12Intensive Care Department, Hospital Virgen de la Concha, Av. Requejo, 35, 49022 Zamora, Spain; dianadonaire@gmail.com; 13Intensive Care Department, Hospital Regional Universitario Carlos Haya, Av. de Carlos Haya, 84, 29010 Málaga, Spain; jf.mtnez88@gmail.com; 14Intensive Care Department, Hospital de Manacor, Carretera Manacor Alcudia, s/n, 07500 Manacor, Spain; rousgastaldi@hotmail.com; 15Intensive Care Department, Hospital General Universitario de Castellón, Avda. de Benicàssim, 128, 12004 Castelló de la Plana, Spain; mateuli@hotmail.com; 16Intensive Care Department, Hospital Universitario de Burgos, Av. Islas Baleares, 3, 09006 Burgos, Spain; mariagero.esc@gmail.com; 17Intensive Care Department, Hospital Universitario de Fuenlabrada, Cam. del Molino, 2, 28942 Fuenlabrada, Spain; b.nietom@gmail.com (B.N.-M.); clara.vaquerizo@salud.madrid.org (C.V.-A.); 18Intensive Care Department, Hospital Universitari Arnau de Vilanova, Av. Alcalde Rovira Roure, 80, 25198 Lleida, Spain; jtruji9@gmail.com (J.T.-C.); lserviag@gmail.com (L.S.-G.); 19IRBLLeida (Institut de Recerca Biomèdica de Lleida Fundació Dr. Pifarré; Lleida Biomedical Research Institute’s Dr. Pifarré Foundation), Av. Alcalde Rovira Roure, 80, 25198 Lleida, Spain

**Keywords:** obesity, medical nutrition therapy, clinical practice guidelines, caloric intake, protein intake, critically ill patients

## Abstract

Background: Critically ill patients with obesity (PwO) have anthropometric characteristics that can be associated with different nutritional-metabolic requirements than other critically ill patients. However, recommendations regarding nutrition delivery in PwO are not clearly established among the different published clinical practice guidelines (CPGs). Our main aim was to evaluate the impact of energy and protein intake in critically ill PwO. Methods: A multicenter (n = 37) prospective observational study was performed. Adult patients requiring medical nutrition therapy (MNT) were included, and PwO (BMI ≥ 30 Kg·m^−2^) were analyzed. Demographic data, comorbidities, nutritional status, and the average caloric and protein delivery administered in the first 14 days, including complications and outcomes, were recorded in a database. Patients were classified and analyzed based on the adequacy of energy and protein intake according to CPG recommendations. Results: 525 patients were included, of whom 150 (28.6%) had obesity. The energy delivery was considered inadequate (<11 Kcal/Kg/d) in 30.7% (n = 46) and adequate (≥11 Kcal/Kg/d) in 69.3% (n = 104) of cases. PwO who received adequate energy delivery had greater use of the parenteral route and longer mean hospital stays (28.6 ± 26.1 vs. 39.3 ± 28.1; *p* = 0.01) but lower ICU mortality (32.6% vs. 16.5%; *p* = 0.02). Protein delivery was inadequate (<0.8 g/Kg/d) in 63.3% (n = 95), insufficient (0.8–1.2 g/Kg/d) in 31.33% (n = 47), and adequate (≥1.2 g/Kg/d) in only 5.4% (n = 8) of patients. PwO with inadequate protein delivery—compared with insufficient delivery—had higher use of the parenteral route and lower mortality in the ICU (25.5% vs. 14.9%; *p* = 0.02). Multivariate analysis revealed that PwO who received adequate energy delivery (hazard ratio [HR]: 0.398; 95% confidence interval [CI]: 0.180–0.882; *p* = 0.023) had better survival, while patients with insufficient protein delivery (HR: 0.404; CI 95%: 0.171–0.955; *p* = 0.038) had better survival than those with inadequate delivery. Conclusion: PwO can frequently receive inadequate energy and protein delivery from MNT during an ICU stay, which may impact the short-term mortality of these critically ill patients. It is emerging to develop strategies to optimize MNT delivery in these patients, which may improve their outcomes. NCT Registry: 03634943.

## 1. Introduction

Appropriate medical nutrition therapy (MNT) is crucial to fulfill the metabolic demands of critically ill patients; however, optimal energy and protein requirements remain a subject of debate [1,2]. Body composition regulates metabolism, whereas critical illness increases catabolism and energy demands, leading to sarcopenia and negative energy balance, which ultimately may influence outcomes [3,4]. As a result, those patients with obesity (PwO) who are critically ill have anthropometric characteristics that may result in different nutritional and metabolic requirements compared with other critically ill patients.

Obesity rates are increasing in Western countries, and consequently, the presence of PwO in the intensive care unit (ICU) is becoming more frequent, with rates reaching up to a third of admitted patients in most ICUs [5,6]. PwO may often present with sarcopenia and malnutrition upon ICU admission due to low physical activity and poor nutrient intake quality [7]. In conjunction with their anthropometric characteristics, they represent a challenge in the ICU to guarantee appropriate macronutrient provision through MNT.

Recommendations regarding nutrition delivery in ICU PwO have not been clearly established among the different published clinical practice guidelines (CPGs) [8]. The American Society of Parenteral and Enteral Nutrition (ASPEN) recommends an energy provision of 11–14 Kcal/Kg/day if the body mass index (BMI) is between 30–50 Kg/m^2^ based on actual body weight. However, the European Society for Clinical Nutrition and Metabolism (ESPEN) recommends an energy provision of 20–25 Kcal/Kg/day for PwO based on adjusted body weight. The same applies to protein requirements: ASPEN recommends 2–2.5 g protein/Kg/day based on ideal body weight, whereas ESPEN recommends 1.3 g protein/Kg/day based on adjusted body weight [8,9]. These differences in energy and protein provision between CPG recommendations, as well as basing them on actual, ideal, or adjusted body weight, may be explained by findings from studies in ICU PwO measuring resting energy expenditure (REE) and comparing differences between indirect calorimetry and several formulas. Fatty tissue has minimal metabolic activity and represents 20–40% of the body composition of PwO, whereas REE increases with BMI, which leads the experts to recommend 60–70% of REE to avoid the deleterious effects of overnutrition (e.g., hyperglycemia) in these patients [10,11,12,13]. In addition, nutrition targets from CPGs are frequently difficult to achieve in current ICU clinical practice, where there is often a gap between what is really delivered to the patients in terms of macronutrient intake (i.e., energy and protein) and the recommended amount of MNT [4,14].

The present research provides further analysis of MNT delivery in critically ill PwO, particularly regarding energy and protein delivery. The main objective was to evaluate the impact of energy and protein intake in critically ill PwO. We also evaluated the adequacy of those intakes based on current CPG recommendations and nutrition delivery in PwO compared with other BMI subgroups (i.e., normal and overweight patients).

## 2. Materials and Methods

### 2.1. Design and Setting

A nationwide prospective observational multicenter study involving 37 Spanish ICUs was performed. Eligible ICUs had to follow a nutritional protocol consistent with current clinical guidelines or involve medical personnel specialized in artificial nutrition therapy [14]. MNT management, including the route of MNT administration, was determined daily by the attending intensivist based on the patient’s clinical status.

These results correspond to a planned post hoc analysis of the insights obtained from PwO (BMI ≥ 30 kg·m^−2^) in the ENPIC study. All consecutive patients admitted between 1 April and 31 July 2018 were included in the study. Patients admitted to participating ICUs, aged 18 years or older, who required MNT (i.e., enteral nutrition, parenteral nutrition, or both) for >48 h and with an expected stay of >72 h were eligible. Patients in the ICU for postoperative recovery from surgery and/or ICU monitoring who did not require specific organ support treatment (e.g., vasopressors or non-invasive mechanical ventilation [non-IMV]) and those who were able to feed orally were excluded from the study. These patients were excluded as they were less severe, had a good prognosis, and had a very low risk of developing gastrointestinal dysfunction; they did not usually need MNT, and their inclusion could have biased the results, affording a more benign picture of what MNT represents in the critically ill [15]. PwO who required ICU admission were included in the analysis of the present study.

The Clinical Research Ethics Committee of Hospital Universitari de Bellvitge—a central institutional review board—approved the study (code PR401/17). Due to the study’s observational design and the use of data from an anonymous centralized database, the requirement for informed consent was waived. Patients included in this study were obtained from the Evaluation of Nutritional Practices in the Critical Care registry (ENPIC Study; ClinicalTrials.gov Identifier: NCT 03634943).

### 2.2. Data Collection and Study Endpoints

The following data were collected for each patient: demographics; BMI; comorbidities (e.g., hypertension, diabetes mellitus, chronic obstructive pulmonary disease, acute myocardial infarction, chronic renal failure, immunosuppression, and active cancer); type of patient (medical, trauma, surgery); Acute Physiology and Chronic Health Evaluation (APACHE) II score; Simplified Acute Physiology Score (SAPS) II score; Sequential Organ Failure Assessment (SOFA) score at ICU admission; nutrition status using Subjective Global Assessment (SGA) and modified Nutrition Risk in the Critically Il (mNUTRIC) scores; details of nutritional therapy, including time of MNT initiation, early MNT within 48 h of ICU admission, and energy and protein intake throughout nutrition therapy administration—or at least the first 14 days; and ICU outcomes, including the need for and duration of IMV, vasoactive drug support, renal replacement therapy (RRT), respiratory tract infection, catheter-related bloodstream infection, length of ICU stay, and ICU and 28-day mortality rates. We also registered enteral nutrition (EN)-related complications. High gastric residual volume (GRV) was defined as an aspirated volume from the stomach of ≥500 mL via a nasogastric tube following EN administration [16]. Aspirations were conducted at each clinician’s discretion based on the observational nature of the present study.

To evaluate nutrition delivery in PwO admitted to the ICU, we compared energy and protein delivery according to different CPG recommendations [8,9]. To this end, PwO were categorized into subgroups based on energy and protein delivery according to the ASPEN and ESPEN recommendations. Thus, we calculated mean energy delivery based on actual body weight (i.e., following ASPEN recommendations) and mean protein delivery based on adjusted body weight (i.e., following ESPEN recommendations). This categorization based was on the preliminary exploratory analysis: when we categorized PwO based on CPG recommendations, only 25 patients (16.6% of PwO) fulfilled energy delivery from ESPEN recommendations (i.e., at least 20 Kcal/Kg/day based on adjusted body weight), and none of the patients fulfilled protein delivery regarding ASPEN recommendations (i.e., at least 2 g protein/Kg/day based on ideal body weight).

When comparing PwO receiving different energy and protein deliveries, study endpoints were differences in outcomes during the ICU stay (i.e., mechanical ventilation needs, vasoactive drug support, RRT, respiratory and catheter-related infections, length of ICU and hospital stays, ICU and 28-day mortality rates).

A STROBE checklist (Strengthening the Reporting of Observational Studies in Epidemiology) was performed to evaluate and ensure a structured reporting method and assess the reproducibility, transparency, and validity of the present research (provided in Appendix A) [17].

### 2.3. Statistical Analysis

Categorical variables are expressed as frequencies and percentages, and continuous variables as mean ± standard deviation (SD). The chi-square test was used to compare the distribution of categorical variables between study groups, while the two-sample *t*-test or Mann–Whitney U test was applied to continuous variables. Differences in characteristics and outcomes between BMI subgroups were analyzed using ANOVA, followed by a post hoc Bonferroni test to identify significant pairwise differences. Statistical significance was set at *p* < 0.05 (two-tailed). Missing data were handled by excluding some of the patients from the analyses (as shown in Figure 1A) since our database did not have a significant amount of missing data (≈5%).

Subsequent multivariate analysis using an adjusted multiple stepwise Cox regression analysis was performed to add a time perspective. Variables deemed suitable by the investigators based on careful consideration of confounding and with a *p*-value < 0.2 were included in the initial model. Investigators selected variables based on current knowledge and the literature. We used the change-in-estimates criterion and backward deletion with a 10% cutoff to eliminate variables from the final model. To avoid destabilizing the multivariate analyses, we tested for interactions between all variables introduced.

We adjusted for age, patient type (e.g., medical, surgical, or trauma), illness severity (e.g., APACHE score), length of nutritional therapy, and data presenting significant differences in baseline characteristics (e.g., sex, comorbidities) between subgroups. This helped avoid confounders and the influence of illness severity when analyzing outcomes. Hazard ratios (HRs) and 95% confidence intervals (CIs) were estimated [18].

Statistical analysis was performed with using PASW statistics 20.0 (SPSS Inc., Chicago, IL, USA). In all cases, the Kolmogorov–Smirnov and D’Agostino–Pearson omnibus normality tests were used to check the normal distribution of our population and assess the goodness-of-fit of the final regression models. No data transformations were necessary as all variables included in the multivariate analysis met the normality assumption.

## 3. Results

### 3.1. Study Population

A total of 644 ICU patients received MNT during the study period and fulfilled the study-inclusion criteria. However, 15 patients were excluded due to insufficient or incomplete data. From the remaining 629, a further 87 patients who only received MNT during 72 h were excluded, as well as 17 patients who were underweight (i.e., BMI < 18.5 kg·m^−2^).

Figure 1 shows the patient population flow chart. The remaining 525 patients were divided according to BMI groups: normal (18.5–24.9 Kg·m^−2^; 31%, n = 165), overweight (25–29.9 Kg·m^−2^, 40%, n = 210), and obese (≥30 Kg·m^−2^; 29%, n = 150) (Figure 1B). It is important to remark that none of the patients had a BMI ≥ 50 Kg·m^−2^.

### 3.2. Population Characteristics of Patients with Obesity Compared with Other BMI Subgroups

General demographics and comorbidities, MNT characteristics, and outcomes of the entire population and all subgroups based on BMI classification are shown in Table 1 (summarized data) and Appendix A. PwO—as well as other BMI subgroups and the entire patient cohort—were admitted due to medical and major surgical diseases, and one-third of PwO presented with malnutrition based on the SGA scale when evaluated upon ICU admission, especially when compared with the normal BMI subgroup. PwO exhibited a higher incidence of cardiovascular risk factors (i.e., hypertension and diabetes mellitus) than the other BMI subgroups, the proportion of males was lower than in the overweight subgroup, and they had a higher incidence of previous acute myocardial infarction than the normal BMI subgroup. Upon ICU admission, PwO presented similar illness severity based on ICU prognosis scores (SOFA, APACHE II, SAPS II) (Table 1 and Appendix A).

### 3.3. Nutrition Delivery and Outcomes in Patients with Obesity Compared with Other BMI Subgroups

In our study, a mean delivery > 70% of recommended energy and protein requirements (i.e., 20 Kcal·Kg^−1^·day^−1^ and 1.2–1.3 g protein) was achieved in 89.7% (n = 148) and 76.9% (n = 127), respectively, for the normal BMI subgroup. However, the overweight subgroup showed similar data regarding energy but not protein delivery, achieving a mean energy delivery > 70% of recommended requirements in 81.4% (n = 171) of patients, whereas only 54.3% (n = 114) achieved a mean protein delivery > 70% of recommended requirements (*p* = 0.03). Finally, PwO only achieved a mean energy and protein delivery > 70% of recommended requirements in 62.6% (n = 94) and 36.3% (n = 55) of patients, respectively (*p* = 0.001).

Most patients received early MNT (i.e., within the first 48 h of ICU admission), regardless of BMI subgroup. However, PwO tended toward later MNT initiation and exhibited significantly lower mean energy (based on actual body weight) and protein delivery (adjusted body weight) compared with the overweight and normal BMI subgroups (Table 1 and Appendix A). The rate of EN-related complications was similar among subgroups.

Once adjusted by confounding factors, multivariate analysis confirmed that PwO received a lower mean amount of protein compared with normal (HR 0.72; 95% CI 0.650–0.850; *p* = 0.007) and overweight patients (HR 0.86; 95% 0.72–0.96; *p* = 0.01).

Regarding ICU outcomes, despite the lack of statistically significant differences, PwO tended toward a higher need for IMV, RRT, and longer ICU stay compared with the other subgroups.

### 3.4. Assessment of Nutrition Therapy in Patients with Obesity: Energy Delivery

When we categorized the mean energy delivery of PwO based on ASPEN recommendations (i.e., 11–14 Kcal/Kg/day based on actual body weight), 69.3% (n = 104) received a mean caloric delivery of ≥11 Kcal/Kg/d and 30.7% (n = 46) received <11 Kcal/Kg/d during the entire MNT administration or at least during the first 14 days (Table 2 and Appendix A).

The subgroup of patients who received adequate energy delivery (i.e., ≥11 Kcal/Kg/d) in comparison with those who did not (i.e., <11 Kcal/Kg/d) had a higher incidence of neoplastic comorbidities, tended to be older and with a higher incidence of malnutrition upon ICU admission (based on SGA), but with less organ failure (i.e., lower SOFA score).

We found that patients who received adequate energy delivery more frequently used the parenteral route. Regarding outcomes, they had a higher mean length of ICU and hospital stays but lower mortality. They also showed a trend toward higher mean days on IMV.

Multivariate analysis, adjusted for confounding factors such as age, patient type (e.g., medical, surgical, or trauma), illness severity, nutritional status (based on SGA), length of MNT, and other data exhibiting significant differences in baseline characteristics, showed that adequate caloric delivery (i.e., ≥11 Kcal/Kg/d) was associated with lower mortality (Table 3A).

### 3.5. Assessment of Nutrition Therapy in Patients with Obesity: Protein Delivery

When we categorized protein delivery of PwO based on ESPEN recommendations (i.e., 1.3 g protein/Kg/day based on adjusted body weight), 63.3% (n = 95) received a mean protein delivery of <0.8 g/Kg/day (inadequate), 31.3% (n = 47) received between 0.8 and <1.3 g/Kg/day (insufficient), and only 5.4% (n = 8) received ≥1.3 g/Kg/d (adequate) throughout MNT administration or at least during the first 14 days.

We only compared differences between inadequate and insufficient protein delivery due to the low number of patients who received adequate protein delivery (Table 4 and Appendix A). Patients who received insufficient protein delivery were older and had a higher incidence of neoplastic comorbidities. They also tended toward a higher incidence of malnutrition upon ICU admission (based on SGA) and a higher degree of illness severity (based on APACHE and SOFA scores).

MNT was more frequently administered via the parenteral route in the insufficient protein delivery subgroup. Regarding outcomes, the insufficient delivery subgroup showed lower mortality in comparison with the inadequate subgroup, and a trend toward longer ICU stays and higher mean days on IMV.

Multivariate analysis was adjusted for confounding factors (see Section 2.3: statistical analysis), such as age, patient type (e.g., medical, surgical, or trauma), illness severity, nutritional status (based on SGA), length of MNT, and other data with significant differences in baseline characteristics. The analysis showed that insufficient protein delivery (0.8–<1.3 g/Kg/day) was associated with lower mortality compared with inadequate protein delivery (<0.8 g/Kg/day).

## 4. Discussion

This observational research focused on analyzing nutrition practices in PwO admitted to the ICU, particularly regarding energy and protein delivery throughout MNT. In our population, PwO presented lower mean energy and protein delivery during ICU admission when compared with other BMI subgroups. Most importantly, we also showed that an adequate mean energy (i.e., ≥11 Kcal/Kg/d) and insufficient protein delivery (≥0.8–<1.3 g/Kg/day)—the latter when compared with inadequate mean protein delivery (<0.8 g/Kg/day)—were both associated with lower ICU mortality rates in PwO. We believe our results help highlight the importance of MNT in the outcomes of critically ill PwO admitted to the ICU.

We found no delay in initiating MNT or greater illness severity in PwO compared with other BMI subgroups—both factors that could be associated with gastrointestinal dysfunction and may ultimately explain the lower energy and protein-delivery targets in these patients [7,19]. In clinical practice, there are several factors complicating MNT administration in critically ill PwO, such as the presence of adiposity—which may lead to an inaccurate abdominal exam to monitor the tolerance of MNT—the underrecognition of malnutrition and sarcopenia, and altered nutrient processing and pharmacokinetics [8,20]. All these factors, together with the anthropometric characteristics of PwO, could negatively influence MNT delivery in general and, more specifically, EN delivery [8].

When we analyzed subgroup differences regarding delivery adequacy and outcomes, we found that PwO, compared with other BMI subgroups, have better nutritional status upon ICU admission (based on SGA). This may be linked to the perception of a greater nutritional reserve in these patients, leading to the misidentification of nutritional needs and a delay in the progressive delivery of energy and protein requirements [7]. We also showed that the incidence of neoplasia was higher in those PwO who received better energy and protein delivery, which may be explained by the contemporary major importance that nutrition therapy has in patients with primary or related oncological problems admitted to the ICU [21].

It is not surprising that, as observed in our patient cohort, a high proportion of patients received energy and protein intakes below CPG recommendations, a trend also reported in contemporary studies (i.e., mean protein delivery between 0.8–1.1 g·Kg^−1^·day^−1^) [22,23,24,25]. PwO exhibit wide variations in macronutrient metabolism, which makes nutrition management complex. However, they may respond better to nutrition therapy delivery and benefit from adequate energy delivery, which is supported by our findings [8,26]. An adequate energy delivery, coupled with protein delivery, is required to overcome anabolic resistance (e.g., insulin resistance) and avoid worsening of skeletal muscle atrophy, which can ultimately lead to ICU-related complications and a more difficult recovery from ICU admission [27,28].

Recent trials evaluating the use of protein intakes above recommendations (i.e., >1.5 g/Kg/d) reported negative results [29]. An adequate protein delivery—or at least an initial mean protein delivery between 0.8 and 1.3 g/Kg/day—is associated with positive clinical results, such as lower mortality, in observational studies [4,22]. Indeed, protein delivery is, together with physiotherapy, one of the pillars for avoiding or minimizing sarcopenia, which is of particular importance in PwO considering their higher incidence of sarcopenia before ICU admission and the deleterious clinical consequences derived from it (e.g., longer hospital stays and poor quality of life) [30,31].

We agree with other published reports that progressive energy and protein delivery during the acute phase of critical illness is needed [32]. Nonetheless, our results suggest that an energy and protein delivery below CPG recommendations may negatively impact the outcomes of critically ill PwO.

Finally, in our study, we demonstrated that PN was frequently used in PwO to optimize MNT delivery, achieving better energy and protein delivery. Considering the inherent difficulties that PwO may have in achieving appropriate enteral delivery, the use of PN seems an option to consider when evaluating the route of nutrition therapy in these patients [23,24]. It is important to consider that, in our study, the MNT route was chosen by the attending physician at the ICU based on the observational nature of the study [14].

### Limitations of the Study

The observational design, participant heterogeneity, and lack of strong conclusions should be considered limitations of the present research. Indeed, we would have liked to analyze MNT in extreme obesity (i.e., BMI ≥ 50 Kg·m^−2^) for a better comprehension of the impact of body composition on nutrition therapy. However, none of the patients included in our database fulfilled this anthropometric characteristic. We would also have liked to assess long-term outcomes and recovery metrics, such as muscle strength and quality of life. However, this was not possible due to the lack of data about caloric and protein delivery during the entire ICU admission. It is also important to remark that this study was designed to associate outcomes with nutrition therapy delivery, not to discover the reasons underlying the different degrees of nutrient delivery among PwO (e.g., differences due to clinical barriers, protocol adherence, etc.).

Nevertheless, the present data reflect a real-world practice of MNT delivered to critically ill adult PwO. The multicenter nature, the significant number of patients included, and the optimal statistical performance, which aimed to minimize the influence of confounders (described in the Methods section), are among the strengths of this study. The sample size could be appropriate for the entire ICU population. However, it may not be optimal for subgroup analysis based on the number of PwO analyzed. Our results agree with previous findings in the literature, where increased energy and protein delivery have been associated with improved clinical outcomes in critically ill patients in different BMI subgroups [33,34].

Future research should study the impact of anthropometric characteristics on nutrition therapy and outcomes in critically ill PwO in greater depth to address gaps in current knowledge. First, research on anthropometric composition should be associated with body composition analyses. Second, it would be helpful to promote subanalyses of outcomes in PwO in large trials and even combine data from different trials. Finally, clinical research on nutrition therapy should be associated with the simultaneous evaluation of quality indicators to investigate the reasons behind suboptimal nutrient delivery [8,35,36].

## 5. Conclusions

In the present observational cohort study regarding MNT during ICU admission of critically ill patients with obesity, an approach closer to CPG recommendations of energy and protein delivery from MNT may positively impact the outcome of these patients. Nutritional strategies and future research should be developed and focused on optimizing MNT for PwO in the ICU with the objective of translating optimal MNT management into a clinical benefit.

## Figures and Tables

**Figure 1 nutrients-17-00732-f001:**
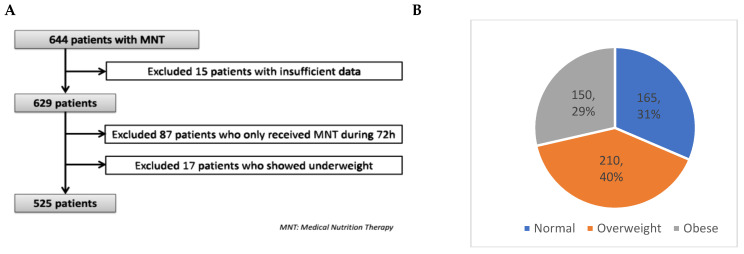
Study flow chart of patients included in the study (**A**) and BMI groups (**B**).

**Table 1 nutrients-17-00732-t001:** General characteristics, nutritional therapy, and outcomes of patients admitted to the ICU based on the body mass index subgroup.

		All Patients n = 525	Normal n = 165	Overweight n = 210	Obese n = 150	*p*-Value
**Baseline Characteristics & Comorbidities**						
Age, years, mean ± SD		61.5 ± 15	58.8 ± 16.5	62.8 ± 14.7	62.7 ± 13.5	0.05
Gender, male patients, n (%)		67.2% (353)	64.8% (107)	74.8% (157)	59.3% (89)	**0.003 ^B^**
Hypertension, n (%)		43.6% (229)	33.9% (56)	41.9% (88)	56.7% (85)	**0.01 ^A,B^**
Diabetes mellitus, n (%)		25% (131)	21.2% (35)	20% (42)	36% (54)	**0.001 ^A,B^**
AMI, n (%)		14.1% (74)	8.5% (14)	16.7% (35)	16.7% (25)	**0.04 ^B^**
Neoplasia, n (%)		20.6% (108)	24.2% (40)	19.5% (41)	18% (27)	0.11
Type of patient	Medical, n (%)	63.8% (335)	65.5% (108)	62.9% (132)	63.3% (95)	0.81
	Trauma, n (%)	12.6% (66)	10.9% (18)	15.2% (32)	10.7% (16)	0.75
	Surgery, n (%)	23.6% (124)	23.6% (39)	21.9% (46)	26% (39)	0.67
**Prognosis ICU Scores and Nutrition Status Upon ICU Admission**						
APACHE II, mean ± SD		20.3 ± 7.9	19.7 ± 7.6	20.1 ± 7.5	21.2 ± 8.5	0.18
SOFA, mean ± SD		7.2 ± 3.4	6.8 ± 3.5	7.2 ± 3.3	7.6 ± 3.5	0.54
Malnutrition (based on SGA), n (%)		41% (215)	52.7% (87)	37.1% (78)	33.3% (50)	**0.01 ^B^**
**Characteristics of Medical Nutrition Therapy**						
Early MNT, <48 h, n (%)		74.9% (393)	77.6% (128)	75.2% (158)	71.3% (107)	0.43
Kcal/kg/day *, mean ± SD		19 ± 5.6	23.1 ± 6	18.6 ± 3.7	15.27 ± 4.24	**0.001 ^A^**
Protein, g/kg/day *, mean ± SD		1 ± 0.4	1.2 ± 0.4	1 ± 0.3	0.8 ± 0.2	**0.01 ^A,B^**
EN		63.2% (332)	59.4% (98)	64.3% (135)	66% (99)	0.34
PN		15.4% (81)	13.3% (22)	16.2% (34)	16.7% (25)	0.85
EN-PN		7.8% (41)	8.5% (14)	7.6% (16)	7.3% (11)	0.92
PN-EN		13.5% (71)	18.8% (31)	11.9% (25)	10% (15)	0.27
**EN-Related Complications**						
Any complication		23.2% (122)	20.6% (34)	23.8% (50)	25.3% (38)	0.12
High GRV		12.4% (65)	11.5% (19)	13.3% (28)	12% (18)	0.54
Diarrhea		9% (47)	6.7% (11)	11.4% (24)	8% (12)	0.18
**Outcomes**						
Mechanical ventilation, n (%)		92.8% (487)	89.1% (147)	93.8% (197)	95.3% (143)	0.08
Mechanical ventilation, days, mean ± SD		15.1 ± 16	13.5 ± 12.3	15.1 ± 13.7	16.7 ± 21.4	0.09
Vasoactive drug support, n (%)		77% (404)	73.9% (122)	79.5% (167)	76.7% (115)	0.44
Renal replacement therapy, n (%)		16.6% (87)	16.4% (27)	12.9% (27)	22% (33)	0.07
ICU stay, days, mean ± SD		20.3 ± 18	18.2 ± 13.8	21.1 ± 17.1	21.6 ± 22.5	0.08
Hospital stay, days, mean ± SD		39.1 ± 32.5	40.9 ± 39	39.8 ± 29.7	36 ± 27.8	0.18
ICU mortality, n (%)		24.4% (128)	24.8% (41)	26.2% (55)	21.3% (32)	0.73
28-day mortality, n (%)		26.7% (140)	29.1% (48)	27.1% (57)	23.3% (35)	0.51

AMI: Acute myocardial infarction; PN: Parenteral Nutrition; EN: Enteral Nutrition; SD: standard deviation; APACHE II: Acute Physiology and Chronic Health Disease Classification System II; SOFA: Sequential Organ Failure Assessment; SGA: Subjective Global Assessment; GRV: Gastric residual volume; ICU: Intensive Care Unit. * During the entire duration of nutrition therapy or at least the first 14 days. Statistically significant *p*-values are written in bold. Statistical results correspond to ANOVA *p*-values. Bonferroni post hoc testing with statistically significant differences: ^A^ between the normal weight and obese subgroups; and ^B^ between the overweight and obese subgroups.

**Table 2 nutrients-17-00732-t002:** General characteristics, nutritional therapy, and outcomes of patients with obesity admitted to the ICU based on mean energy delivery categories.

		All Obese n = 150	<11 Kcal/Kg/d n = 46	≥11 Kcal/Kg/d n = 104	*p*-Value
**Baseline Characteristics and Comorbidities**					
Age, years, mean ± SD		62.7 ± 13.5	60.07 ± 12.79	63.84 ± 13.7	0.05
Gender, male patients, n (%)		59.3% (89)	63.0% (29)	57.7% (60)	0.66
Hypertension, n (%)		56.7% (85)	54.3% (25)	57.7% (60)	0.84
Diabetes mellitus, n (%)		36% (54)	36.9% (17)	35.6% (37)	0.99
Neoplasia, n (%)		18% (27)	6.5% (3)	23.1% (24)	**0.03**
Type of patient	Medical, n (%)	63.3% (95)	60.9% (28)	64.4% (67)	0.81
	Trauma, n (%)	10.7% (16)	17.4% (8)	7.7% (8)	0.09
	Surgery, n (%)	26% (39)	21.7% (10)	27.8% (29)	0.55
**Prognosis ICU Scores and Nutrition Status Upon ICU Admission**					
APACHE II, mean ± SD		21.2 ± 8.5	22.2 ± 8.8	20.8 ± 8.4	0.18
SOFA, mean ± SD		7.6 ± 3.5	8.2 ± 3.7	7.3 ± 3.4	0.08
Malnutrition (based on SGA), n (%)		33.3% (50)	28.9% (13)	35.6% (37)	0.23
**Characteristics of Medical Nutrition Therapy**					
Early MNT, <48 h, n (%)		71.3% (107)	65.2% (30)	74.1% (77)	0.36
EN		66% (99)	84.8% (39)	57.7% (60)	**0.002**
PN		16.7% (25)	4.3% (2)	22.1% (23)	0.01
EN-PN		7.3% (11)	8.7% (4)	6.7% (7)	0.74
PN-EN		10% (15)	2.2% (1)	13.5% (14)	**0.04**
**EN-related Complications**					
Any complication		25.3% (38)	30.4% (14)	23.1% (24)	0.08
High GRV		12% (18)	15.2% (7)	10.6% (11)	0.14
**Outcomes**					
Mechanical ventilation, n (%)		95.3% (143)	100% (46)	93.3% (97)	0.17
Mechanical ventilation, days, mean ± SD		16.7 ± 21.4	13.41 ± 17.23	18.25 ± 23.09	0.08
Vasoactive drug support, n (%)		76.7% (115)	71.7% (33)	78.8% (82)	0.46
Renal replacement therapy, n (%)		22% (33)	15.2% (7)	25% (26)	0.18
ICU stay, days, mean ± SD		21.6 ± 22.5	17.1 ± 19.3	23.7 ± 23.6	**0.04**
Hospital stay, days, mean ± SD		36 ± 27.8	28.6 ± 26.1	39.3 ± 28.1	**0.01**
ICU mortality, n (%)		21.3% (32)	32.6% (15)	16.5% (17)	**0.02**
28-day mortality, n (%)		23.3% (35)	34.8% (16)	18.7% (19)	**0.03**

PN: Parenteral Nutrition; EN: Enteral Nutrition; SD: standard deviation; APACHE II: Acute Physiology and Chronic Health Disease Classification System II; SOFA: Sequential Organ Failure Assessment; SGA: Subjective Global Assessment; GRV: Gastric residual volume; ICU: Intensive Care Unit. Statistically significant *p*-values are written in bold.

**Table 3 nutrients-17-00732-t003:** Results of multivariate analyses of factors associated with energy (**A**) and protein (**B**) delivery categories of patients with obesity during the ICU stay.

Variables	Hazard Ratio (95% Confidence Interval)	*p*-Value
A- Dependent variable: energy delivery ≥ 11 Kcal/Kg/d		
Neoplasia	1.209 (0.901–1.975)	0.45
Days on mechanical ventilation	2.595 (0.505–1.887)	0.07
ICU mortality	0.398 (0.180–0.882)	**0.023**
B- Dependent variable: protein delivery ≥ 0.8–<1.3 g/Kg/day		
Mechanical ventilation	0.990 (0.915–1.605)	0.67
ICU mortality	0.404 (0.171–0.955)	**0.038**

Quality of the model: Akaike Information Criterion (AIC) = 186.03; Bayesian Information Criterion (BIC) = 205.4; Pseudo R2 (McFadden) = 0.21. Statistically significant *p*-values are written in bold.

**Table 4 nutrients-17-00732-t004:** General characteristics, nutritional therapy, and outcomes of patients with obesity admitted to the ICU based on mean protein delivery categories.

		All Obese n = 150	<0.8 g/Kg/day n = 95	≥0.8–<1.3 g/Kg/day n = 47	≥1.3 g/Kg/d n = 8	*p*-Value **
**Baseline Characteristics and Comorbidities**						
Age, years, mean ± SD		62.7 ± 13.5	60.1 ± 13.1	67 ± 12.5	67.5 ± 17.8	**0.01**
Gender, male patients, n (%)		59.3% (89)	57.9% (55)	65.9% (31)	37.5% (3)	0.28
Hypertension, n (%)		56.7% (85)	53.7% (51)	59.6% (28)	75% (6)	0.48
Diabetes mellitus, n (%)		36% (54)	34.7% (33)	38.3% (18)	37.5% (3)	0.91
Neoplasia, n (%)		18% (27)	10.5% (10)	31.9% (15)	25% (2)	**0.004**
Type of patient	Medical, n (%)	63.3% (95)	64.2% (61)	61.7% (29)	62.5% (5)	0.96
	Trauma, n (%)	10.7% (16)	13.7% (13)	6.4% (3)	0	0.36
	Surgery, n (%)	26% (39)	22.1% (21)	31.9% (15)	37.5% (3)	0.36
**Prognosis ICU Scores and Nutrition Status Upon ICU Admission**						
APACHE II, mean ± SD		21.2 ± 8.5	21.1 ± 8.3	22.6 ± 8.5	14.8 ± 8.9	0.14
SAPS II, mean ± SD		52 ± 18.7	50 ± 17.6	57 ± 20.4	45.6 ± 17	0.18
SOFA, mean ± SD		7.6 ± 3.5	7.5 ± 3.5	8.1 ± 3.5	5.3 ± 3.4	0.12
Malnutrition (based on SGA), n (%)		33.3% (50)	29.8% (28)	36.2% (17)	62.5% (5)	0.16
**Characteristics of Medical Nutrition Therapy**						
Early MNT, <48 h, n (%)		71.3% (107)	72.6% (69)	68.1% (32)	75% (6)	0.90
EN		66% (99)	75.8% (72)	53.2% (25)	25% (2)	**0.001**
PN		16.7% (25)	12.6% (12)	19.1% (9)	50% (4)	**0.03**
EN-PN		7.3% (11)	6.3% (6)	10.6% (5)	0	0.64
PN-EN		10% (15)	5.2% (5)	17% (8)	25% (2)	**0.02**
**EN-Related Complications**						
Any complication		25.3% (38)	26.3% (25)	27.6% (13)	50% (4)	0.36
High GRV		12% (18)	7.4% (7)	12.7% (7)	50% (4)	0.08
**Outcomes**						
Mechanical ventilation, n (%)		95.3% (143)	100% (95)	91.5% (43)	62.5% (5)	0.45
Mechanical ventilation, days, mean ± SD		16.7 ± 21.4	14.5 ± 16.7	22.3 ± 29.4	7.8 ± 6.6	0.09
Vasoactive drug support, n (%)		76.7% (115)	73.7% (70)	80.8% (38)	87.5% (7)	0.60
Renal replacement therapy, n (%)		22% (33)	22.1% (21)	23.4% (11)	12.5% (1)	0.94
ICU stay, days, mean ± SD		21.6 ± 22.5	19.5 ± 17.8	27.6 ± 30.4	12.4 ± 5.1	0.06
Hospital stay, days, mean ± SD		36 ± 27.8	33 ± 27.6	40.8 ± 26.1	43.4 ± 38.7	0.23
ICU mortality, n (%)		21.3% (32)	25.5% (24)	14.9% (7)	12.5% (1)	**0.02**
28-day mortality, n (%)		23.3% (35)	25.2% (24)	21.3% (10)	12.5% (1)	0.07

PN: Parenteral Nutrition; EN: Enteral Nutrition; SD: standard deviation; APACHE II: Acute Physiology and Chronic Health Disease Classification System II; SOFA: Sequential Organ Failure Assessment; SGA: Subjective Global Assessment; GRV: Gastric residual volume; ICU: Intensive Care Unit. ** Statistically significant differences between inadequate (i.e., <0.8 g/Kg/day) and insufficient (0.8–<1.3 g/Kg/day) protein delivery subgroups. Statistically significant *p*-values are written in bold.

## Data Availability

The data presented in this study is available via contact with the corresponding author of the present manuscript. Data requests should be evaluated by the local ethics committee to comply with legal requirements.

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
