# Peer review of "Nutrition Therapy in Critically Ill Patients with Obesity: An Observational Study"

_nutrients, 2025, doi:10.3390/nu17040732_

Round 1
Reviewer 1 Report
Comments and Suggestions for Authors
Thank you for the opportunity to review this valuable work. I have no objections to the ethical issues of the study, it received approval from the bioethics committee. Particularly noteworthy is the multicenter nature of this study and its nature.
Here are some comments that I suggest you take into consideration
1. Please consider changing the phrase obese with the phrase patients with obesity throughout the text. This is more of a linguistic and political issue.
2. Line 109 Please remove the words It is important to remark that
3. Line 377 - Please consider removing the words s. We think that
4. Line144. isn't it worthwhile to add an interval in which one aspires 500 ml ?
5. Line 162 - Is it possible to check what catecholamines were administered, in what dose range?
6. The discussion is very broad and well prepared
7. The selection of literature is correct, while the conclusions are based solely on the analyses carried out
Author Response
Responses to the reviewer are in the attached document.

Reviewer 2 Report
Comments and Suggestions for Authors I would like to thank the authors for their manuscript. The manuscript explores the critical topic of nutrition management in obese ICU patients, focusing on energy and protein delivery through medical nutrition therapy. Comments: 1. Introduction section: - The introduction provides relevant background information and establishes the importance of the study. However, the rationale for focusing specifically on critically ill obese patients could be elaborated further to strengthen the argument. - The opening sentence, “Despite appropriate medical nutrition therapy (MNT) is crucial for the metabolic demands of critically ill patients,” is slightly awkward. Consider revising for grammatical clarity, e.g., “Although appropriate medical nutrition therapy (MNT) is crucial for meeting the metabolic demands of critically ill patients, the optimal energy and protein requirements remain a subject of debate. - Briefly explain the basis of the recommendations by ASPEN and ESPEN (e.g., the reasoning behind using actual, adjusted, or ideal body weight) to add depth to the introduction. - The study objectives are clear, but the phrasing could be refined. For instance: This study aims to analyse MNT delivery in critically ill obese patients, with a focus on energy and protein intake.” - Consider making the second objective more specific 2. Methods section: - The eligibility criteria are clear, but the rationale for excluding patients admitted for postoperative recovery or without organ support needs further justification. For example, explain how their inclusion might have confounded the results. - Clarify how missing data were handled during analysis (e.g., imputation methods, exclusion). - The use of normality tests (Kolmogorov–Smirnov and D’Agostino–Pearson) is appropriate. Specify if any data transformations were applied when assumptions were violated. 3. Results section: - The results section is detailed and presents comprehensive data, but its clarity could be improved by breaking down dense paragraphs into smaller, thematically focused segments. This would help readers navigate the findings more easily. - Results in some parts are reported inconsistently (e.g., mix of percentages and absolute numbers). Standardizing the format throughout the section will enhance readability. - Why obese patients achieved lower targets for energy and protein delivery. Were these differences due to clinical barriers, protocol adherence, or other factors? - In the adjusted multivariate analyses , please explain how confounding factors were selected and why these particular variables were included in the model. - This section contains a lot of results interpretations that may be better suited for the discussion section. - The tables in the results section are overly crowded and include a significant amount of information, much of which is not subsequently discussed in the discussion section. The authors should consider summarizing the most critical and relevant findings in a more concise format. Non-essential data could be included as supplementary material if deemed necessary for reference. 4. The discussion section: - The discussion attempts to address multiple aspects of nutrition practices in obese ICU patients but lacks a clear focus on the main findings. The authors should organize the discussion to prioritize key results and their implications, followed by secondary considerations. - There are several important findings (e.g., lower energy and protein delivery in obese patients, subgroup differences in delivery adequacy, and outcomes) that are briefly mentioned but not fully explored. The authors should elaborate on these findings. - Some points are repeated unnecessarily (e.g., the role of PN and challenges in EN). Consolidating these ideas would make the discussion more concise. - Highlight unique findings and how they address gaps in current knowledge, avoiding overreliance on general challenges of MNT in obese patients. - Detail how limitations may have impacted specific results, offering suggestions for future research to address these gaps.
Author Response
Responses to the reviewer are in the document attached.

Reviewer 3 Report
Comments and Suggestions for Authors
This paper addresses a significant gap in the field of medical nutrition therapy for critically ill obese patients. The study highlights a lack of clarity and consensus in existing guidelines for the nutritional needs of critically ill obese patients. Based on the methodology outlined in the article, here are potential improvements and additional controls the authors might consider:
1.Ensure that BMI subgroups are matched for key variables such as age, sex, comorbidities, and illness severity to eliminate potential confounders.
2.A broader inclusion of patients with extreme obesity (BMI ≥50 kg/m²) could provide a more comprehensive dataset.
3.Extend the follow-up period beyond 14 days to assess long-term outcomes, including recovery metrics such as muscle strength and quality of life.
These suggestions aim to refine the methodology, enhance the reliability of findings, and address potential biases or confounders.
Author Response

(The authors gave the same response as above.)

Reviewer 4 Report
Comments and Suggestions for Authors
Dear Authors,
First of all, I would like to express my sincere gratitude for giving me the opportunity to contribute my opinion in the evaluation of your manuscript. I found the topic discussed extremely interesting and relevant to the field in which we operate. The research presents numerous useful and promising insights that could lead to significant advancements in our sector. However, after a thorough reading, I believe there are some aspects, mainly methodological, that need to be improved and clarified in order to allow for a full appreciation of the proposed work. Below, I outline the main areas that could benefit from further elaboration and revision.
Editing:
I believe that the tables presented are not very user-friendly due to their size; I suggest synthesizing the crucial elements of the study in the text and including the full versions of the tables as supplementary files.
Title:
Perhaps "delivery" could be misleading; I suggest using the more traditional term "support."
Abstract:
There is a lack of a real interpretation in the conclusions regarding the clinical practice implications of the phenomenon studied, along with potential suggestions and specific proposals for the relevant scientific community. Furthermore, the registration number is included in the conclusions, which I believe is a typo.
Keywords:
Okay.
Introduction:
I believe this section could be improved with a broader epidemiological approach to the phenomenon studied, and further expanded, perhaps supported by an extended bibliography.
Methods:
This is the most controversial aspect, which certainly deserves more attention, namely the absence of a structured reporting method, such as the STROBE checklist (doi:10.1016/j.jclinepi.2007.11.008), which could have been used to ensure the scientific validity and transparency of the study. Including this checklist as a supplementary file, along with its citation in the text, would be fundamental to improving the quality and reproducibility of the study, making it more transparent and scientifically valid.
Results:
This is certainly the strength of the study. I would just pay attention to the appropriate use of acronyms and make the tables as user-friendly as possible, as suggested in the editing section.
Discussion:
I suggest reviewing the section related to possible comparisons with other healthcare settings, such as other European or American countries, which are more closely comparable to the patients considered in the study.
Limits:
In my humble opinion, it would be helpful to create a specific section for limitations but ensure it has the correct numbering (4.1 limitations).
Bibliography:
It should be expanded according to the previous suggestions, and perhaps updated for references older than 15-20 years, unless they are of methodological type or have significant impact evidence.
In summary, the manuscript presents scientifically valuable results, but requires several methodological and structural improvements to enhance its overall quality. My advice is to proceed with a thorough revision addressing these points before moving forward with publication, as, if appropriately modified, the manuscript could represent a significant contribution to the relevant scientific literature.
Author Response

(The authors gave the same response as above.)

Round 2
Reviewer 4 Report
Comments and Suggestions for Authors
Dear Authors,
very good job. Please only change sex in gender.
Best.
Author Response
Thank you so much for your comment and dedicated work. We have changed sex in gender by your request.